# Genome-Wide Identification of *WRKY* Gene Family and Expression Analysis under Abiotic Stress in Barley

**Junjun Zheng, Ziling Zhang, Tao Tong, Yunxia Fang, Xian Zhang, Chunyu Niu, Jia Li, Yuhuan Wu, Dawei Xue** 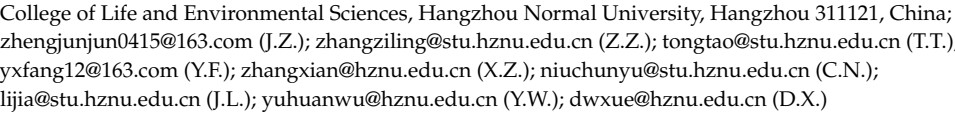
**and Xiaoqin Zhang \***

College of Life and Environmental Sciences, Hangzhou Normal University, Hangzhou 311121, China; zhengjunjun0415@163.com (J.Z.); zhangziling@stu.hznu.edu.cn (Z.Z.); tongtao@stu.hznu.edu.cn (T.T.); yxfang12@163.com (Y.F.); zhangxian@hznu.edu.cn (X.Z.); niuchunyu@stu.hznu.edu.cn (C.N.); lijia@stu.hznu.edu.cn (J.L.); yuhuanwu@hznu.edu.cn (Y.W.); dwxue@hznu.edu.cn (D.X.)
**\*** Correspondence: zxq@hznu.edu.cn; Tel.: +86-571-28860639

**Abstract:** The *WRKY* gene family consists of transcription factors that are widely distributed in plants and play a key role in plant growth and development, secondary metabolite synthesis, biotic and abiotic stress responses, and other biological processes. In this study, 86 WRKY proteins were identified from the barley genome database using bioinformatics and were found to be distributed unevenly on seven chromosomes. According to the structure and phylogenetic relationships, the proteins could be classified into three groups and seven subgroups. The multiple sequence alignment results showed that WRKY domains had different conserved sites in different groups or subgroups, and some members had a special heptapeptide motif. Protein and gene structure analysis indicated that there were significant differences between the groups in terms of the distribution of WRKY motifs and the number of introns in barley. Tissue expression pattern analysis demonstrated that the transcription levels of most genes exhibited tissue and growth-stage specificity. In addition, the analysis of *cis*-elements in the promoter region revealed that almost all *HvWRKYs* had plant hormone or stress response *cis*-elements, and there were differences in the numbers between groups. Finally, the transcriptional levels of 15 *HvWRKY* genes under drought, cadmium, or salt stress were analyzed by quantitative real-time PCR. It was found that most of the gene expression levels responded to one or more abiotic stresses. These results provide a foundation for further analysis of the function of *WRKY* gene family members in abiotic stress.

**Keywords:** abiotic stress response; barley; genome-wide identification; tissue-specific expression; WRKY

## 1. Introduction

The WRKY proteins belong to a class of transcription factors that contain special domains. They are called WRKY proteins due to their conserved WRKYGQK amino acid core sequences in the N-terminal of their characteristic domains [1]. They all regulate gene transcription by binding to the conserved W-box in the promoter region of the target gene. WRKY proteins can be divided into three groups based on their sequence characteristics: group I contains two WRKY domains, and group II and group III both contain one WRKY domain. However, group II has a C-terminal zinc-finger structure of C2H2, and group III has a zinc-finger structure of C2HC. Based on phylogenetic relationships, group II proteins can be further divided into five subgroups: II a, II b, II c, II d, and II e [1].

WRKY transcription factors are widespread in plants, participating in many metabolic pathways such as plant growth, development, pathogen tolerance, abiotic stress responses, and senescence [2]. Among these functions, there have been extensive studies in the field of abiotic stress. For example, under cadmium (Cd) stress, *AtWRKY13* gene expression is induced. The overexpression of *AtWRKY13* leads to a decrease in Cd accumulation and the enhancement of Cd tolerance, and the loss of its function will lead to increased Cd accumulation and sensitivity. Furthermore, studies have showed that *AtWRKY13*

can directly bind to the promoter of the *PDR8* gene to activate its transcription, and the *Arabidopsis thaliana* (L.) Heynh. ABC transporter PDR8 is a Cd extrusion pump that confers Cd tolerance in plants, thus positively regulating Cd tolerance in *A. thaliana* [3]. The overexpression of the soybean transcription factor *GmWRKY49* in *A. thaliana* can improve the salt tolerance of plants, and similar results have been observed in soybean composite seedlings overexpressing *GmWRKY49*, confirming that *GmWRKY49* is a positive regulator in the salt tolerance pathway of soybean [4]. In rice, the *OsWRKY11* gene can directly bind to the promoter of *RAB21* to mediate the drought response. The ectopic expression of the *OsWRKY11* gene can enhance the tolerance of plants to drought stress [5]. *ZmWRKY106* can regulate the expression of stress-related genes through the abscisic acid (ABA) signaling pathway. The overexpression of *ZmWRKY106* significantly improved the drought and heat tolerance of *A. thaliana* [6].

*SPF1*, the first member of the *WRKY* gene family, was isolated from sweet potato (*Ipomoea batatas* (L.) Lam.). *SPF1* can recognize the SP8 sequences in the 5′ upstream regions of sporamin and *β*-amylase genes and participate in the establishment of the sugar signaling pathway [7]. Since then, especially with the development of whole-genome sequencing, the *WRKY* gene family has been identified in many plants, such as *A. thaliana,* with more than 70 members [2,8]; rice, with 107 members [9]; soybean, with 182 members [10]; pineapple, with 54 members [11]; tomato, with 81 members [12]; cotton, with 116 members [13]; and *Brachypodium distachyon* (L.) Beauv., with 86 members [14].

Barley (*Hordeum vulgare* L.) is the fourth-largest cereal crop in the world after maize, rice, and wheat. It is one of the oldest food crops and feed crops. The genome of barley has been completely established, which has facilitated widespread molecular biology research in this plant [15,16]. However, there are few reports on the identification of the *WRKY* gene family in barley and its involvement in abiotic stress. In this study, all of the barley WRKY proteins from the genomic database were identified and subjected to phylogenetic analysis, alignment of conserved domains, analysis of gene and protein structure, and *cis*-element analysis of the promoter region. In addition, we analyzed the transcription levels of these genes at different growth stages and in different tissues according to the available data, and Morex barley seedlings were treated with salt, polyethylene glycol (PEG) 6000, and cadmium chloride ($CdCl_2$) stress. The transcription levels of 15 *HvWRKY* genes were determined by quantitative real-time (qRT)-PCR to evaluate the response of the genes to abiotic stress. This study provides a foundation for the identification and further functional analysis of WRKY family members in barley, thus facilitating the cultivation of stress-resistant cultivars and improving the yield and quality of barley.

## 2. Materials and Methods

### 2.1. Identification of WRKY Family Members in Barley and Their Chromosomal Locations

The FASTA files of candidate sequences containing WRKY domain were downloaded from the Leibniz-Institut Für Pflanzengenetik Und Kulturpflanzenforschung IPK Barley BLAST Server (https://webblast.ipk-gatersleben.de/barley_ibsc/downloads/; accessed on 25 February 2019).The hidden Markov model (HMM) based on the *Arabidopsis* WRKY protein downloaded from the TAIR website (https://www.arabidopsis.org/; accessed on 3 March 2019) was used to identify barley WRKY protein in HMMER 3.0 [17] In all, 103 sequences containing the WRKY DNA binding domain (PF03106) were identified as candidate proteins. The redundant sequences were deleted online on the CD-hit website (http://weizhong-lab.ucsd.edu/cdhit-web-server/cgi-bin/index.cgi?cmd=cd-hit; accessed on 3 March 2019) [18], the identity of the cutoff sequence was set to 0.9, and 98 non-redundant sequences were reserved. Then, these sequences were submitted to the Statistical Metabolomics Analysis-An R Tool (SMART) website to confirm whether there were WRKY domains [19]. After removing the sequences without typical WRKY domains, 86 HvWRKY proteins were obtained. The MG2C v2.1 website (http://mg2c.iask.in/mg2c_v2.1/; accessed on 24 June 2019) [20] was used to visualize the distribution of the *WRKY* gene on seven barley chromosomes. TBtools was used to analyze tandem duplication and segmen-

tal duplication of barley *WRKY* genes, as well as the synteny relationship among four plant genomes [21].

## 2.2. Phylogenetic Analysis of WRKY Family Members and Sequence Alignment

The amino acid sequences of WRKY proteins in *Arabidopsis thaliana* and barley were alignment by ClustalX2.1 [22]. Based on the amino acid sequences of the WRKY domain in *Arabidopsis* and barley, the phylogenetic tree of WRKY was constructed by neighbor-joining (NJ) method of MEGA7.0 with 1000 bootstrap replications [23]. Then, visualization and optimization were carried out in iTOL [24]. All WRKY domain sequences of predicted HvWRKY proteins were aligned using DNAMAN software.

## 2.3. Structural Analysis of WRKY Genes

The motifs of all predicted HvWRKY proteins were analyzed by the MEME function of TBtools [19]. The parameters of MEME are as follows: set any number of motifs for each sequence; the maximum motif number is 10; the motif width is between 10 and 50 aa, and the rest parameters are the default values. The gene sequence of each predicted HvWRKY protein was downloaded from the barley genome database. The distribution pattern of intron was analyzed by Gene Structure Display Server (GSDS) (http://gsds.cbi.pku.edu.cn/; accessed on 12 April 2019) [25]. SPSS v20.0 one-way ANOVA was used to analyze the number of introns in each group. The Tukey method was used for pairwise comparison.

## 2.4. Expression Profile of WRKY Genes at Different Growth Stages and in Different Tissues

According to the expression levels of genes from 15 samples at different growth stages and from different tissues provided by the IPK Barley BLAST Server (https://webblast.ipk-gatersleben.de/barley_ibsc/; accessed on 8 March 2019), the expression profiles of 86 genes were drawn by MeV software [26]. The 15 samples were collected from embryos, etiolated seedlings, seedling roots, seedling shoots, epidermal strips, developing inflorescences, inflorescences rachis, roots, third internode of the tillers, lodicule, lemma, palea, developing grains (5 days after pollination), developing grains (15 days after pollination), and senescing leaves. After the expression levels were normalized in MeV, different colors were used to show different expression levels.

The expression patterns of the different genes were clustered by hierarchical clustering, and the genes with different expression patterns were grouped by setting the distance threshold of the gene tree to 4.0. Then, the hierarchical clustering (HCL) tree was obtained to show the relative expression of *HvWRKY* genes at different growth stages and in different tissues.

## 2.5. Stress-Related cis-Elements in HvWRKY Promoter Regions

The upstream sequence (2000 bp) of the *HvWRKY* gene was downloaded from EnsemblPlants (http://plants.ensembl.org/Hordeum_vulgare/Info/Index; accessed on 20 April 2019). Ten stress related *cis*-elements of the *HvWRKY* upstream sequence were identified by PlantCARE (http://bioinformatics.psb.ugent.be/webtools/plantcare/html/; accessed on 27 April 2019) [27], and the distribution of *cis*-elements in each group was statistically analyzed.

## 2.6. Abiotic Stress Treatment at the Barley Seedling Stage

The barley variety Morex was used as the research material. The seeds were spread on wet filter paper and germinated for 48 h at 24 °C in darkness, transferred to 96-well plates and cultured in Hoagland's medium, and then grown in a chamber at 24 °C with a 14/10 h light/dark cycle. The two-leaf-stage seedlings were treated with 200 mM NaCl, 20% (*w/v*) PEG6000, and 50 μM $CdCl_2$ to induce salt, drought, and cadmium stress, respectively. After 24 h of treatment, the shoots were cut off in groups, immersed in liquid nitrogen for rapid freezing, and then transferred for storage at −80 °C for RNA extraction.

*2.7. Quantitative RT-PCR Analysis of the HvWRKY Gene Response to Abiotic Stress*

According to the gene expression characteristics, 15 *HvWRKY* genes were selected and their transcriptional changes under three stress treatments were detected by qRT-PCR. The reference gene was *HvActin* (*HORVU1Hr1G002840*). The primers were designed by Primer Premier v5.0, and the sequence information of all primers is shown in Table S1. The AxyPrep<sup>TM</sup> Multisource Total RNA Miniprep Kit (Axygene, Union City, CA, USA) was used to isolate total RNA. Genomic DNA was digested and RNA was reverse transcribed into cDNA using a Hifair<sup>TM</sup> II 1st Strand cDNA Synthesis Kit (Yeasen, Shanghai, China). The qRT-PCR was performed in a Bio-Rad CFX system using the Hieff® qPCR SYBR Green Master Mix (Yeasen, Shanghai, China). The PCR conditions were 1 min at 95 °C followed by 40 cycles of 10 s at 95 °C, 20 s at 60 °C, and 20 s at 72 °C. The relative expression of the gene was calculated using the $2^{-\Delta\Delta Ct}$ method. The results are the mean $\pm$ standard deviation (SD) of three replicate experiments.

## 3. Results

### 3.1. Identification and Chromosomal Locations of HvWRKYs

In this study, we identified 86 candidate genes containing the WRKY domain by genome-wide analysis and named them *HvWRKY1* to *HvWRKY86* according to their location on the barley chromosome. The physicochemical properties of the family members were analyzed. The results (Table S2) showed that the length of the 86 proteins ranged from 117 (*HvWRKY33*) to 1573 (*HvWRKY64*) amino acids. The predicted molecular weights (MWs) ranged from 13 to 177 kDa, and the isoelectric points (pIs) ranged from 4.73 (*HvWRKY3*) to 10.17 (*HvWRKY53*). The results of chromosome mapping (Figure S1) showed that 82 of the 86 *HvWRKY* genes could be mapped to seven linkage groups of barley. Among them, there were 21 *HvWRKY* genes on chromosome 3, which was the highest, and only five *HvWRKY* genes on chromosome 6, which was the lowest. In addition, four genes could not be specifically mapped to any linkage group.

Five tandem duplications involving 11 *HvWRKY* genes (*HvWRKY9/10/11* on chromosome 1, *HvWRKY30/31* and *HvWRKY37/38* on chromosome 3, *HvWRKY81/82* on chromosome 7, and *HvWRKY84/85* without anchoring chromosome) were identified in barley WRKY family members using MCScanX of TBtools (Figure S1). In addition, 13 pairs of segmentally duplicated genes of the WRKY family were found in the barley genome (Figure 1). The synteny analysis among rice, *B. distachyon*, barley, and *A. thaliana* showed different degrees of correlation (Figure 2). The number of collinear gene pairs among monocots was much greater than that between barley and *A. thaliana*, indicating that the collinearity of WRKY family members between *A. thaliana* and barley was lower.

### 3.2. Phylogenetic Classification of HvWRKY Genes

Based on the number of WRKY domains and the type of zinc-finger motifs, the HvWRKY proteins were divided into three groups. Group I contained two WRKY domains (one at the N-terminal and the other at the C-terminal) and their zinc-finger motifs were C2H2 type. Group II contained a WRKY domain, and the zinc-finger motif pattern was C2H2. Group III had a WRKY domain and a C2HC zinc-finger motif. There were 10, 42, and 34 HvWRKY proteins belonging to group I, group II, and group III, respectively. The zinc-finger structure of the type III protein HvWRKY18 was incomplete.

In order to further classify the barley *WRKY* genes, 86 HvWRKY proteins and 72 domains of *Arabidopsis* WRKY proteins were used to construct a phylogenetic tree (Figure 3). It was found that HvWRKY76 had only one WRKY domain, but its domain was clustered between the C-terminal WRKY domain of the group I protein, and thus HvWRKY76 may be sourced from a complete group I WRKY protein that lost an N-terminal WRKY domain. Therefore, HvWRKY76 is classified as a group I WRKY protein. Group II could be further divided into five subgroups (II a–II e) according to the phylogenetic relationships. Subgroup II a, II b, II c, II d, and II e contained 5, 4, 16, 7, and 10 members, respectively (Figure 3).

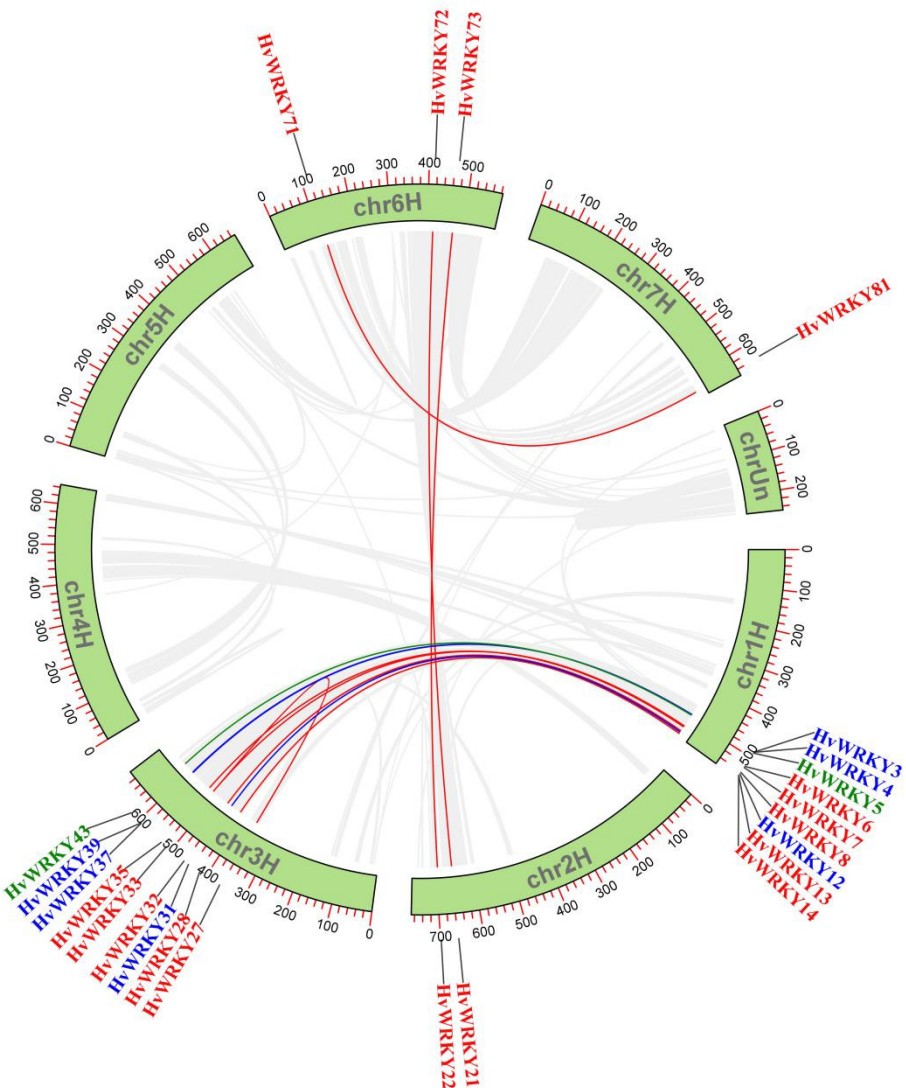

**Figure 1.** The segmental duplication of WRKY family in barley. The Circos map drawn by TBtools shows the position relationship of 12 pairs of segmental duplication gene pairs detected in barley genome. The scale outside the chromosome represents physical position (Mb). The green font is group I WRKY, red is group II, and blue is group III.

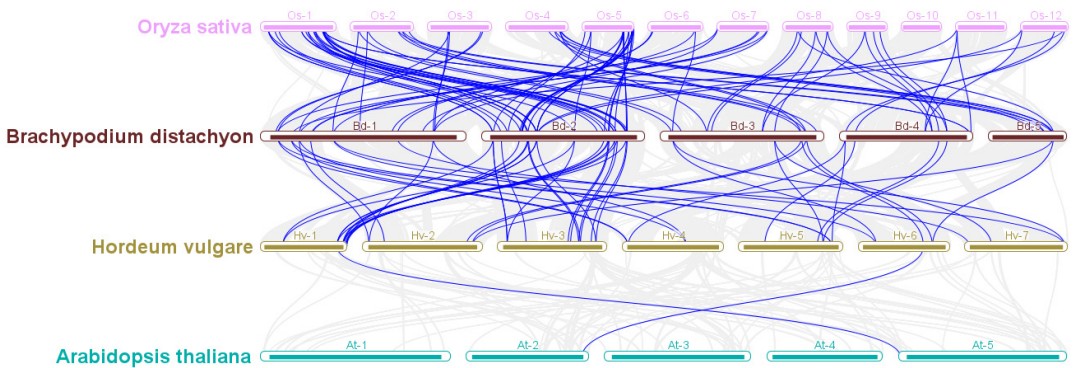

**Figure 2.** Collinearity analysis of *WRKY* genes in *Oryza sativa* L. (pink), *Brachypodium distachyon* (brown), *Hordeum vulgare* (grey), and *Arabidopsis thaliana* (blue). Each horizontal bar represents a chromosome. The orthologous *WRKY* genes were linked using blue curves.

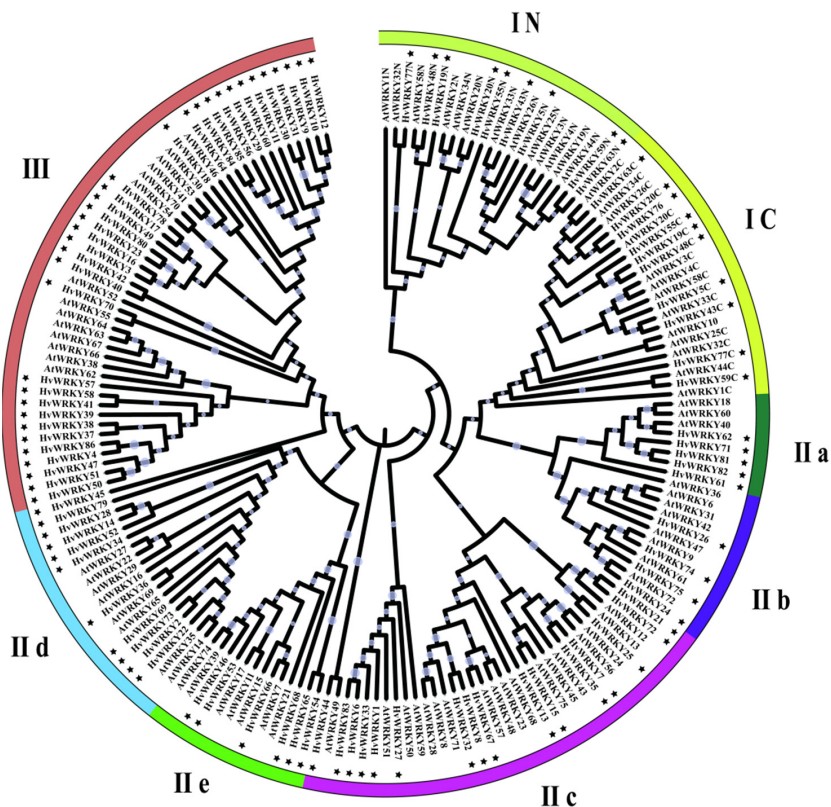

**Figure 3.** Phylogenetic relationship of barley and *Arabidopsis* WRKY proteins. The names of groups and subgroups have been marked outside the ring. "N" and "C" represent the N-terminal and C-terminal WRKY domains of group I members. Asterisks indicate that the WRKY protein comes from barley. The dot size represents bootstrap values. ClustalX is used for multiple sequence alignments of WRKY domain. Phylogenetic tree was constructed by MEGA7.0 using the neighbor-joining (NJ) method with 1000 bootstrap replications.

### 3.3. Multiple Sequence Alignment of WRKY Domains

The multiple sequence alignment results showed that WRKY domains had different conserved sites in different groups or subgroups, especially the zinc-finger structure (Figure 4). All of the WRKY domains from the group I proteins contained a WRKYGQK sequence, and they all possessed a C2H2-type zinc finger (CX3-4CX22-23HXH). A WRKY domain and C2H2 zinc finger (CX4-5CX21-24HXH) were observed in all 42 group-II proteins. Among them, five members of subgroup II c contained a WRKYGKK sequence, and one member of subgroup II d contained a WKKYGQK sequence, while the others all had conservative WRKYGQK sequences. Among the 34 group III proteins contained one WRKY domain, five members contained a WRKYGEK sequence, and two members contained WKKYGQK or WTKYGQK. All of the other members possessed conserved WRKYGQK sequences. With the exception of HvWRKY18, the group III HvWRKYs possessed a C2HC zinc finger (CX3-4CX23-31HXC).

### 3.4. Conserved Motifs of HvWRKY and the Structure of Their Genes

A total of 10 motifs were identified in the barley WRKY proteins by MEME motif analysis (Figure 5). Each HvWRKY possessed a different motif composition. Motif 1 was annotated as a WRKY DNA binding motif (data not shown), which is the basic feature of WRKY transcription factors, with at least one motif 1 in all HvWRKY proteins. Motif 2 was located in the middle of the WRKY domain and exists in most proteins. Motif 3 was found only in group I and subgroup II b/II c proteins. Motifs 4 and 9 appeared in group I/II and group III proteins due to different zinc-finger types. Motif 5/8/10 was found only in a few group III proteins. Motif 6 was mainly distributed in group II a/II b and group III proteins. Motif 7 was found only in group II d and group III proteins.

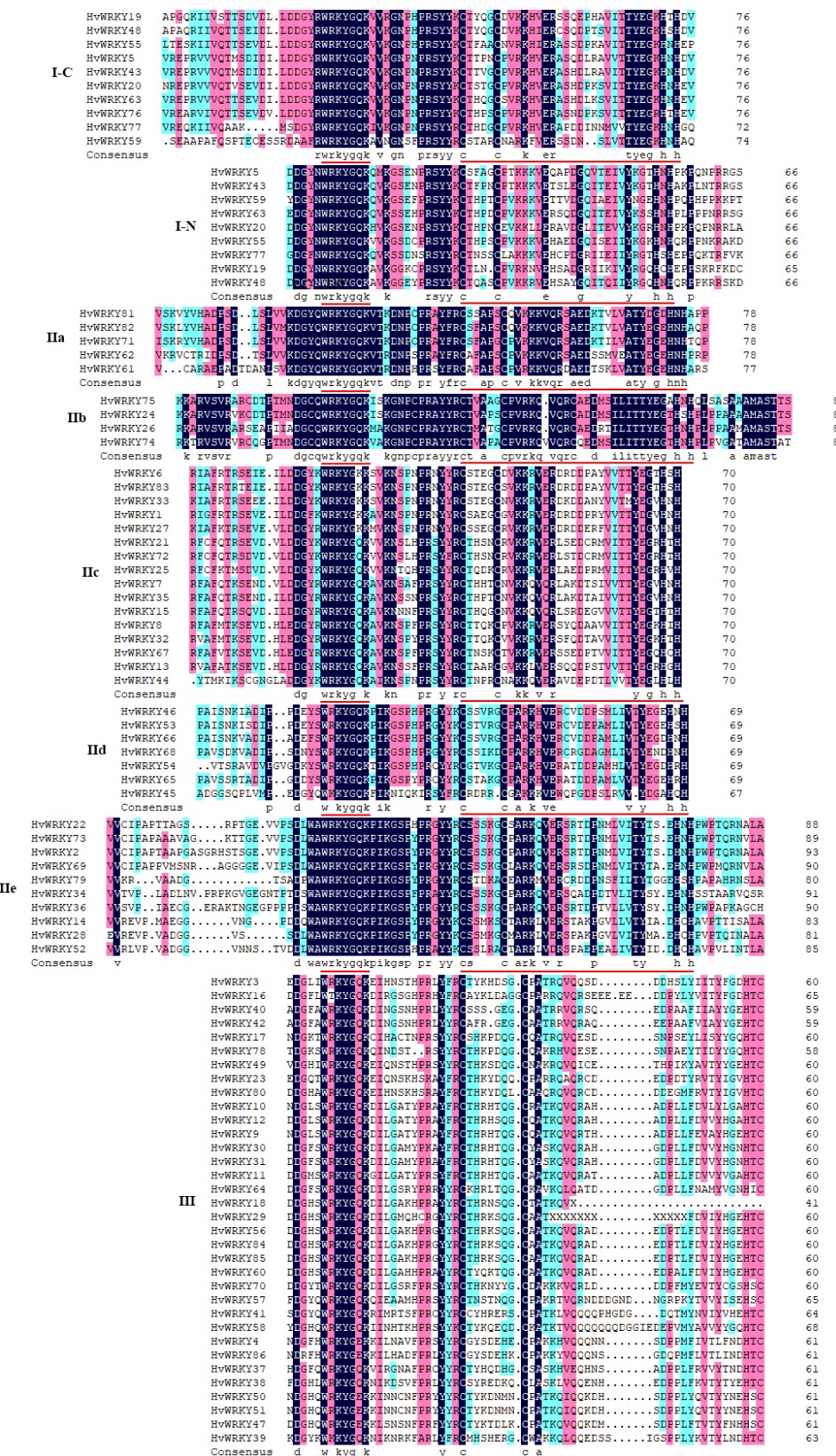

**Figure 4.** Multiple sequence alignments of WRKY domains in each group of HvWRKY proteins. The WRKYGQK core motif and zinc finger structure have been marked with red underline. Each group was aligned on a heptapeptide core motif. Dark blue shading represents a homology level of 100% in a group or subgroup, pink ≥ 75%, and light blue ≥ 50%.

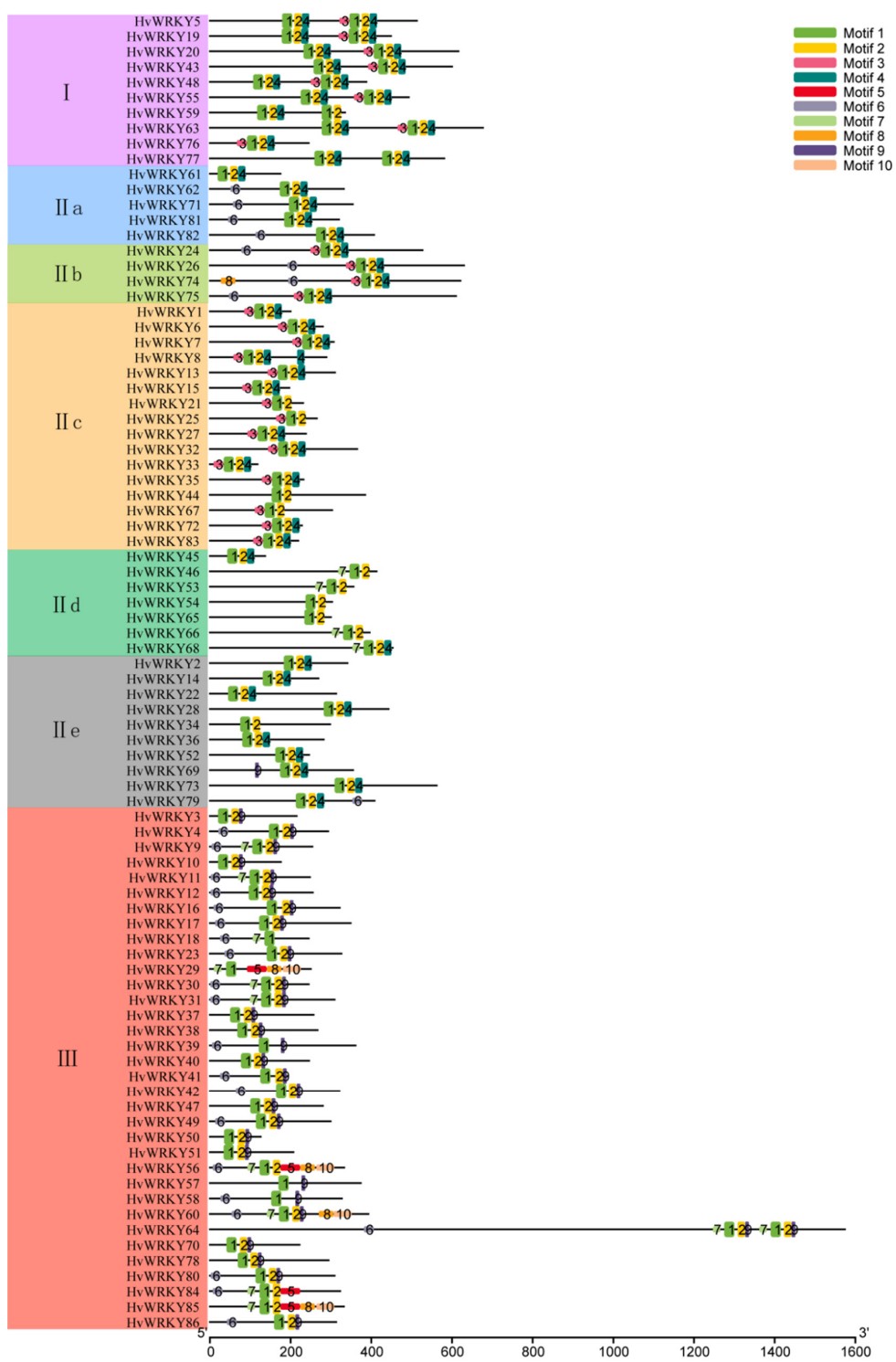

**Figure 5.** Distribution of conserved motifs of HvWRKY protein in each group. Ten motifs were predicted by MEME and are shown in different colors. Motifs 1/2/4/9 were all located in the WRKY domain and represented by a wider rectangle.

Gene structure analysis showed that most *HvWRKYs* had at least one intron, and the number of exons ranged from two to seven, except for seven genes (*HvWRKY1/9/10/11/12/16/54*) (Figure 6). A *HvWRKY* gene with three exons was the most frequent type, accounting for 47.7% of the total. The statistical results of the number of introns in each group showed that the genes encoding group I and subgroup II b proteins had the most introns (Figure S2).

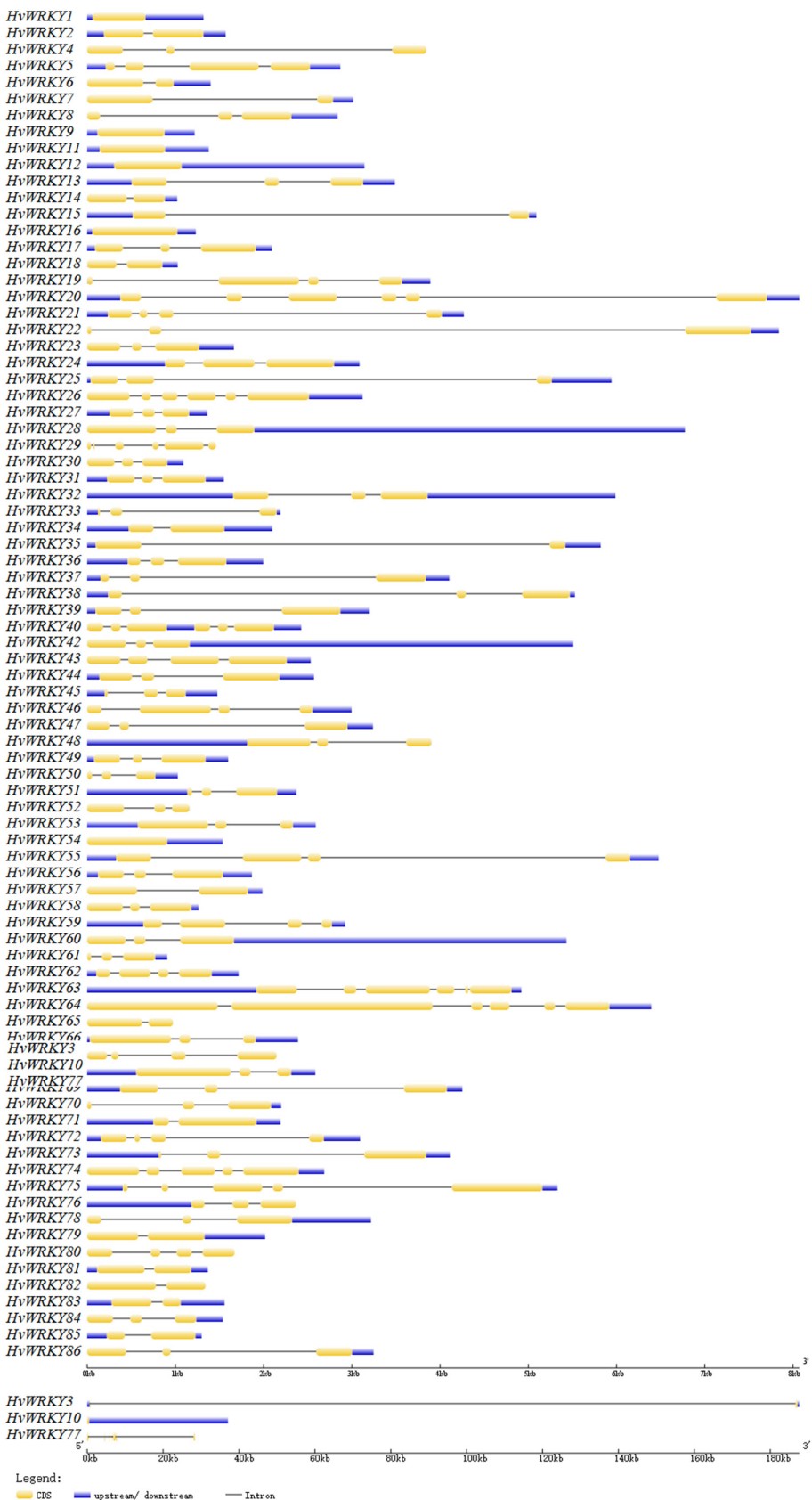

**Figure 6.** Intron–exon structure of *HvWRKY* gene. The gene structure of *HvWRKY* was analyzed by GSDS. The yellow block represents the coding sequence (CDS), the blue block represents the 5' or 3' untranslated region (UTR), and the black line represents intron. Three genes (*HvWRKY3*, *HvWRKY10*, *HvWRKY77*) are displayed on a smaller scale.

### 3.5. Analysis of Expression Levels at Different Growth Stages and in Different Tissues

The expression patterns of genes are often related to their functions, and the analysis of expression patterns can inform functional research. The expression patterns of 86 *HvWRKYs* genes in 15 samples of different tissues at different growth stages were divided into 13 groups, and seven genes did not belong to any group (Figure 7).

Among them, Group 3 contained two *HvWRKY* genes that were highly expressed in young inflorescences; these genes may be involved in inflorescence development. Group 9 contained six *HvWRKY* genes that were highly expressed in senescent leaves, indicating that these genes may be involved in the process of leaf senescence. Group 10 contained 18 *HvWRKY* genes that were highly expressed in the roots, suggesting that these genes may play a role in the roots. Group 11 contained seven *HvWRKY* genes that were highly expressed in the roots of seedlings, indicating that these genes may be involved in the root development of seedlings. Group 13 contained 10 *HvWRKY* genes that were highly expressed in the third internode of the developing tillers, implying that these genes may be involved in the growth and development of stems. In addition, *HvWRKY2* was highly expressed in the inflorescence rachis, indicating that *HvWRKY2* might be involved in the formation of ear traits. The high expression of *HvWRKY44* in the developing grains after five days of flowering indicated that *HvWRKY44* might play a role in the early development of grains.

### 3.6. Analysis of cis-Elements

In order to elucidate the signal transduction pathway of *HvWRKYs* in plants and the regulatory mechanisms during the stress response, the distribution of 10 *cis*-elements in the *HvWRKY* promoter region was analyzed. The 10 stress-response elements included ABRE (ABA response), AuxRR-core (auxin response), CGTCA-motif (MeJA response), LTR (low-temperature response), MBS (MYB transcription factor binding site involved in drought induction), P-box (gibberellin response), TCA-element (salicylic acid response), TC-rich repeats (defense and stress response), TGA-element (auxin-responsive element), and W-box (WRKY transcription factor binding site in the defense response).

In this study, only two genes (*HvWRKY84* and *HvWRKY85*) did not possess any of these *cis*-elements. Elements related to phytohormone regulation were identified in most *HvWRKY* promoters (Figure S3). A total of 74 (86%) *HvWRKYs* had one or more ABREs, suggesting that these genes may respond to ABA. The CGTCA motif involved in the MeJA response also existed in 74 genes. Some *cis*-elements related to stress were also found in some *HvWRKY* promoter regions. For example, there were 46 *HvWRKYs* containing one or more W-box elements in their promoter regions. In addition, LTR, MBS, and TC-rich repeats were found in 41, 38, and eight genes, respectively. According to the *cis*-element statistics of each group (Figure 8), the promoter region of group I member possessed more MBS and less AuxRR-core and TCA-element. There were more ABRE, TCA-element, TC-rich repeats, and W-box and less LTR in the promoter regions of group II members. Group III had more AuxRR-core and TGA-element and less W-box.

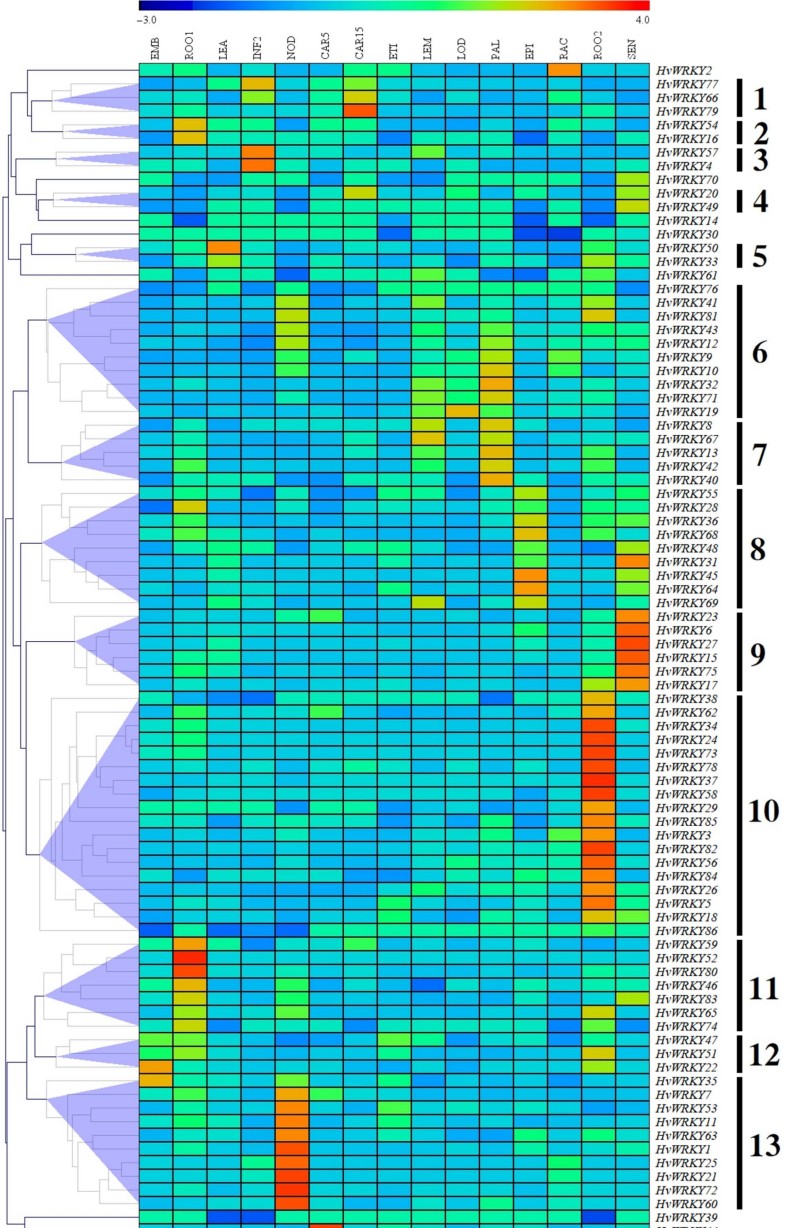

**Figure 7.** The relative transcription level of the *HvWRKY* genes in different growth stages and tissues. Blue represents lower expression, red represents higher expression, and the number on the right represents the groups of expression patterns. The sample sources are as follows: EMB: 4 day embryos; ETI: etiolated seedling, dark condition (10 DAP); ROO1: roots from seedlings (10 cm shoot stage); LEA: shoots from seedlings (10 cm shoot stage); EPI: epidermal strips (28 DAP); INF2: developing inflorescences (1–1.5 cm); RAC: inflorescences, rachis (35 DAP); ROO2: roots (28 DAP); NOD: developing tillers, 3rd internode (42 DAP); LOD: inflorescences, lodicule (42 DAP); LEM: inflorescences, lemma (42 DAP); PAL: dissected inflorescences, palea (42 DAP); CAR5: developing grain (5 DAP), CAR15: developing grain (15 DAP); SEN: senescing leaves (56 DAP).

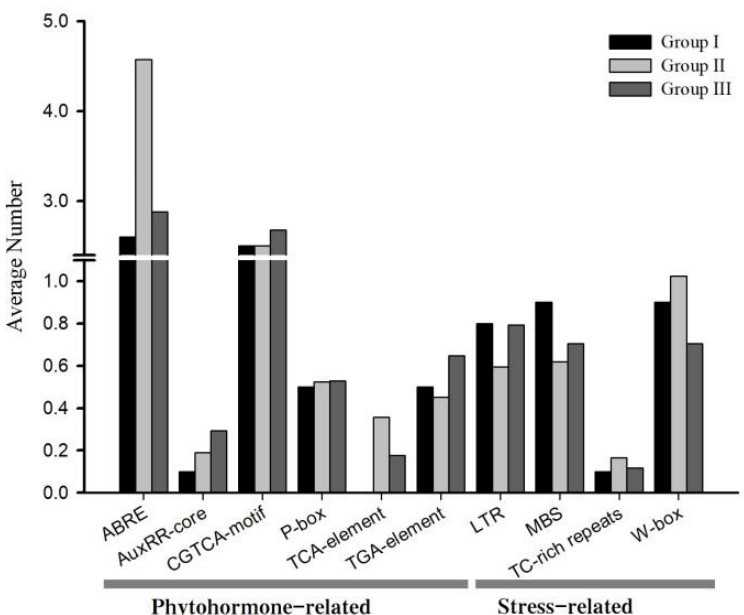

**Figure 8.** Distribution of *cis*-elements in promotor of *HvWRKYs* in each group. Six *cis*-elements on the left side were phytohormone response elements and four elements on the right side were stress related elements.

*3.7. Transcriptional Responses of HvWRKY Genes to Abiotic Stresses*

Many gene families in barley respond to abiotic stresses such as salt, drought, and heavy metals [28,29]. By comprehensively considering the protein subcellular localization, tissue expression level, and protein properties, we selected 15 genes to analyze expression levels under abiotic stress. We used qRT-PCR to analyze the expression profiles of these 15 *HvWRKY* genes under PEG6000, $CdCl_2$, and salt treatment (Figure 9). It can be seen from the figure that most of the 15 *HvWRKY* genes responded to abiotic stress at the transcriptional level. Following PEG6000 treatment, the expression levels of four genes. (*HvWRKY1*, *3*, *24*, *83*) increased by more than twofold, among which the expression of *HvWRKY3* was 13 times higher than that of the control, and the expression of two genes (*HvWRKY32* and *HvWRKY80*) decreased significantly. The expression levels of seven genes (*HvWRKY1*, *3*, *13*, *15*, *24*, *43*, *55*) were increased by more than twofold following treatment with $CdCl_2$. The expression levels of *HvWRKY3* and *HvWRKY24* were 20- and 10-fold higher than those of the control, respectively. The expression levels of two genes (*HvWRKY32* and *HvWRKY80*) decreased significantly, and the expression of *HvWRKY80* decreased to one-fifth of the control. After NaCl treatment, the expression levels of six genes (*HvWRKY1*, *27*, *15*, *43*, *71*, *83*) were increased by more than 1.5 times, and the expression levels of four genes (*HvWRKY3*, *24*, *32*, *80*) were significantly decreased, and the expression levels of *HvWRKY3* and *HvWRKY24* were 1/33 and 1/5 of the control, respectively. Overall, the transcription of four genes (*HvWRKY1*, *13*, *15*, *83*) was upregulated under the three stress treatments. On the contrary, two genes (*HvWRKY32* and *HvWRKY80*) were inhibited. In addition, three genes (*HvWRKY28*, *43*, *71*) were upregulated under salt and Cd stress but were inhibited under simulated drought conditions. Two genes (*HvWRKY3* and *HvWRKY24*) were upregulated under simulated drought and Cd stress but were inhibited under salt stress. *HvWRKY53* was upregulated under drought and Cd stress but was hardly responsive to salt stress. *HvWRKY27* and *HvWRKY55* were upregulated only under Cd stress but hardly responded to the other two stresses.

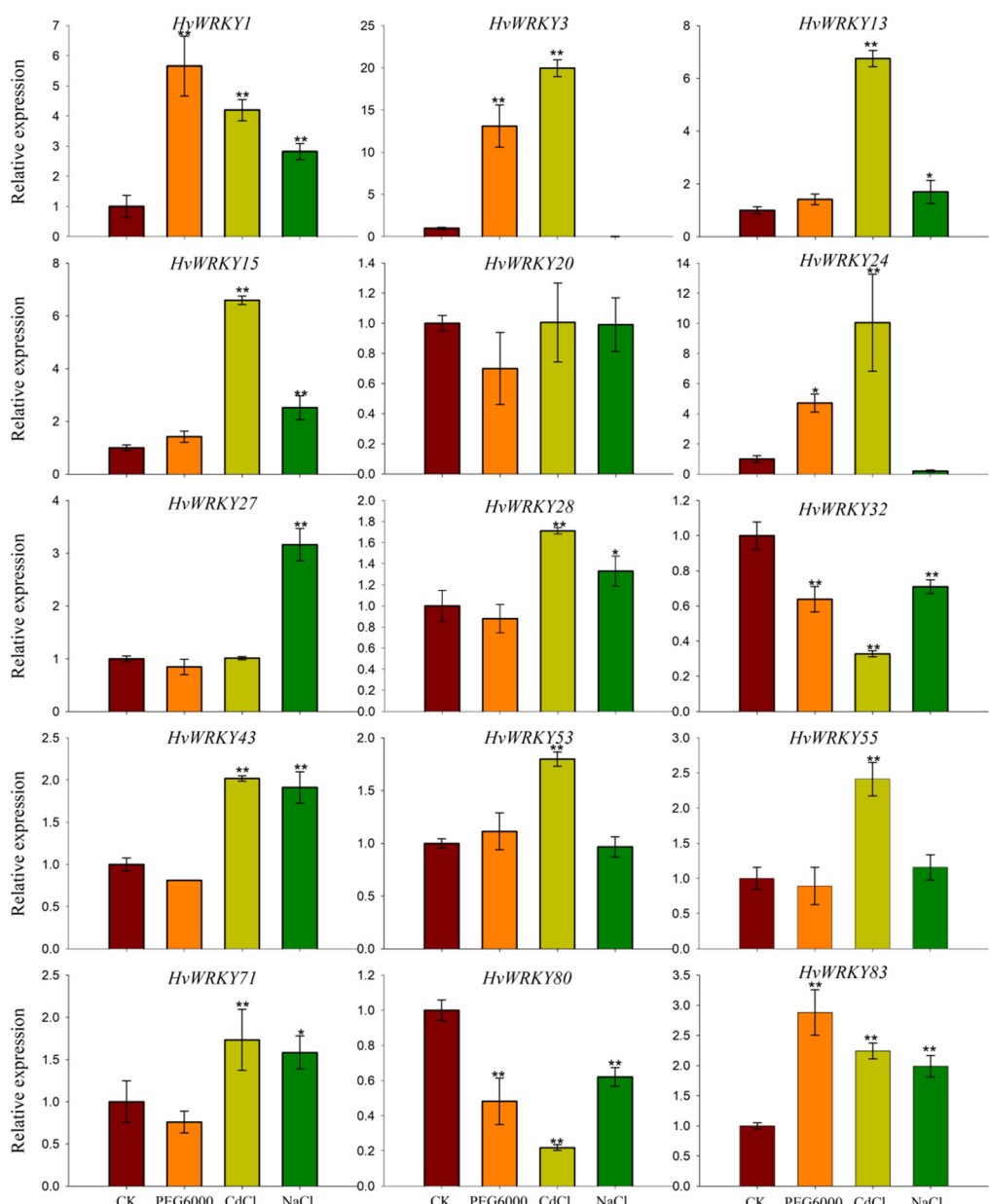

**Figure 9.** Relative expression of 15 *HvWRKY* genes under PEG6000, CdCl$_2$, and NaCl treatments. The data were calculated by the $2^{-\Delta\Delta Ct}$ method. The internal reference gene was *Actin*, and the error line represented the standard deviation (n = 3). Significant differences were determined by one-way ANOVA test: * $p < 0.05$; ** $p < 0.01$.

## 4. Discussion

The *WRKY* gene family exists widely in plant genomes. As more and more plant genomes are being sequenced, the distribution of the WRKY family in the plant kingdom has been increasingly studied [12,30–32]. In this study, we identified 86 WRKY family members in the barley genome. The WRKY domain is the key sequence that determines the specific binding of the WRKY protein to the *cis*-element W-box. The multiple sequence alignment results of the WRKY domain of the HvWRKY proteins showed that the conserved sequences of the WRKYGQK heptapeptide of 13 HvWRKY proteins varied (Figure 4). According to previous studies, this variation may affect the interaction between WRKY proteins and downstream target genes [33,34]. Therefore, these genes should be further studied for their function and binding specificity. In other monocotyledonous plants, such as rice and maize [35–37], the loss and gain of the WRKY domain is common, which is an

important driving force for the expansion of the *WRKY* gene family. In barley, we found that the N-terminal WRKY domain of HvWRKY76 was lost, which was consistent with previous results. It has been reported that the DNA binding activity of the N-terminal WRKY domain is not as strong as that of the C-terminal WRKY domain, and there are more variations in the N-terminal WRKY domain. In accordance with previous studies, the WRKY domains of subgroup II a and II b were clustered together with the C-terminal WRKY domains of group I proteins. These results indicate that the WRKY domains of subgroup II a and II b are more closely related to the C-terminal WRKY domains of group I proteins. Similar to other plants [14,32,38], WRKY also exhibits tandem and segmental duplication in barley, indicating that gene replication plays an important role in the formation of the *WRKY* gene family in barley.

Many studies show that the *WRKY* gene is involved in the regulation of plant growth and development [39,40]. In this study, we analyzed the expression patterns of the *HvWRKY* gene using available transcriptome data at different developmental stages and tissues. Based on the phylogenetic relationships, we obtained some insight into the biological functions of barley *WRKY* genes by comparison with the known functions of *WRKY* genes of *A. thaliana*. *HvWRKY13* clustered with *AtWRKY23* of *A. thaliana*, and *AtWRKY23* can mediate embryo development [41]. In the expression profile, we found that *HvWRKY13* was highly expressed in the lemma and palea, which are near to the seeds, suggesting that this gene is related to embryo or seed development. In addition, *HvWRKY21* and *HvWRKY72* clustered together with *AtWRKY12*, which is expressed in stems and regulates the formation of secondary cell walls [42]. In the expression profile, *HvWRKY21* and *HvWRKY72* were highly expressed in the internodes of the tillers, suggesting that the two genes may have similar functions as *AtWRKY12*. The group III WRKY protein is believed to play an important role in the plant response to stresses [43]. We found that 11 of the 18 Group 10 *HvWRKY* genes that were highly expressed in the roots of seedlings were group III proteins. This proportion (61%) was significantly higher than that of the group III proteins in all of the WRKY family members (39.5%). We speculated that some WRKY group III proteins may play a role in stress-response regulation in barley roots.

Previous studies have shown that subgroup II c WRKY transcription factors can regulate abiotic stress tolerance [44], and some members participate via the ABA signaling pathway [45]. In this study, four subgroup II c genes (*HvWRKY1*, *HvWRKY13*, *HvWRKY15*, and *HvWRKY83*) were all upregulated under simulated drought, Cd, and salt treatments. This suggests that WRKY II c may be involved in the positive regulation of the barley response to these three stresses. There are ABRE elements in the promoter region of these four genes, suggesting that these genes are upregulated through the ABA pathway, which is consistent with previous studies. With the exception of *HvWRKY1*, the other three genes have a W-box in their promoter region, which indicates that these genes may have autoregulation or cross-regulation with other *WRKY* genes. The promoter region of *HvWRKY43* in response to Cd and salt treatments contained the ABRE, CGTCA-motif, and TGA-element at the transcriptional level, indicating that *HvWRKY43* may participate in the response to heavy metal and salt stress through the ABA, MeJA, or auxin signaling pathways.

## 5. Conclusions

Based on the available genomic data, 86 HvWRKY proteins were identified. The genes encoding these proteins were distributed unevenly on seven chromosomes of barley, and there were five tandem duplications and 13 pairs of segmentally duplicated genes. In addition, WRKY family members of rice, *Brachypodium distachyon*, barley, and *Arabidopsis thaliana* showed different number of collinear gene pairs. These 86 proteins could be divided into three groups or seven subgroups according to their structure and phylogenetic relationship, the motif distribution showed obvious group differences, and the sequence differences of WRKY domain also matched our classification. Based on the *cis*-element analysis of the promoter region, we hypothesized that many *HvWRKYs* are involved in

phytohormone signaling and abiotic stress response, and the frequency of some elements was different among three groups. Transcriptional analysis of 15 *HvWRKY* genes under PEG6000, cadmium, and salt treatment revealed that some *HvWRKY* genes responded to one or more abiotic stresses. Our study provides a foundation for further functional research and identification of the barley *WRKY* gene family, thus assisting in elucidating the molecular mechanisms of the barley response to abiotic stress.

**Supplementary Materials:** The following are available online at https://www.mdpi.com/2073-4395/11/3/521/s1, Figure S1: Distribution of *HvWRKY* genes on barley chromosomes, Figure S2: Comparison of the average number of introns in *WRKY* gene family members of each group or subgroup, Figure S3: cis-Elements distribution of *HvWRKYs* promoters, Table S1: The primer sequence used in qRT-PCR, Table S2: Basic information of protein and nucleic acid of *WRKY* gene family members in barley.

**Author Contributions:** Conceptualization, D.X. and X.Z. (Xiaoqin Zhang); methodology, J.Z., T.T. and D.X.; software, J.Z. and Z.Z.; validation, J.Z., T.T. and X.Z. (Xian Zhang); formal analysis, J.Z., Y.F. and C.N.; resources, D.X., Y.W. and X.Z. (Xiaoqin Zhang); data curation, J.Z. and Z.Z.; writing—original draft preparation, J.Z., C.N. and J.L.; writing—review and editing, Y.F., D.X. and X.Z. (Xiaoqin Zhang). All authors have read and agreed to the published version of the manuscript.

**Funding:** This research was funded by the National Natural Science Foundation of China (31401316) and the Hangzhou Scientific and Technological Program (20140432B03).

**Institutional Review Board Statement:** Not applicable.

**Informed Consent Statement:** Not applicable.

**Data Availability Statement:** Not applicable.

**Conflicts of Interest:** The authors declare no conflict of interest.

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
