# Peer review of "Genome-Wide Identification of WRKY Gene Family and Expression Analysis under Abiotic Stress in Barley"

_agronomy, doi:10.3390/agronomy11030521_

Round 1

Reviewer 1 Report

Zheng et al, performed the WRKY gene family analysis in barley genome. For their manuscript, they identified 86 WRKYs distributing unevenly on 7 chromosomes. Structure and phylogenetic relationships indicated the proteins were classified into different groups. Tissue expression pattern analysis demonstrated that most genes exhibited tissue-specific expression. The analysis of cis-elements in the promoter region revealed that almost all HvWRKYs had plant hormone or stress response cis-elements with different numbers of members of gene family. The expression of 15 HvWRKY genes under drought, cadmium, or salt stress were validated by quantitative real-time PCR revealed that most of the gene expression levels responded to one or more abiotic stresses.  For the analysis of WRKY gene family in plants, these analyses are enough, but som descriptions need to be fixed.

  1. Authors mentioned in abstract that “The multiple sequence alignment results showed that some of the WRKY domains were mutated.” I did not find this analysis in manuscript. Should be added in manuscript. Except that, from MSA analysis, we could not get the mutation conclusion among different members of WRKY gene family, only get the difference among them. This must be addressed in revised manuscript.
  2. this sentence “These results indicated that gene duplication, particularly segmental duplication, was 170 the main driving force for the evolution of the WRKY family in barley.” In page 4, line 170 should be removed, or authors should supply more clear data support.
  3. p4, l173-174 this sentence should be revised. MCScan analysis could supply collinear analysis, but its could not supply the analysis of the expansion of gene family.
  4. authors should check the comparison of abstract and conclusion in manuscript, which should be have logical relationship.

Author Response

1. Authors mentioned in abstract that “The multiple sequence alignment results showed that some of the WRKY domains were mutated.” I did not find this analysis in manuscript. Should be added in manuscript. Except that, from MSA analysis, we could not get the mutation conclusion among different members of WRKY gene family, only get the difference among them. This must be addressed in revised manuscript.

Response: Thank you for your suggestion. We have corrected the summary of the MSA analysis in the abstract to “The multiple sequence alignment results showed that WRKY domains had different conserved sites in different groups or subgroups, and some members have special heptopeptide motif.” And the corresponding adjustment is made in the "Result".

2. this sentence “These results indicated that gene duplication, particularly segmental duplication, was the main driving force for the evolution of the WRKY family in barley.” In page 4, line 170 should be removed, or authors should supply more clear data support.

Response: Thanks. We have deleted the "These results indicated that gene duplication, particularly segmental duplication, was the main driving force for the evolution of the WRKY family in barley." in paragraph 2 of 3.1.

3. p4, l173-174 this sentence should be revised. MCScan analysis could supply collinear analysis, but its could not supply the analysis of the expansion of gene family.

Response: Thanks. We have changed the last sentence of the second paragraph of 3.1 to "The number of collinear gene pairs among monocots was much greater than that between barley and A. thaliana, indicating that the collinearity of WRKY family members between A. thaliana and barley was lower.”

4. authors should check the comparison of abstract and conclusion in manuscript, which should be have logical relationship.

Response: Thanks. We have rewritten the 'Conclusions' so that they are more about presenting the results of the study as a whole, rather than highlighting the functional assumptions of certain groups or family members. In addition, the modified conclusion has a closer logical relationship with the abstract.

Reviewer 2 Report

In the current study the authors performed a genome-wide analysis of the WRKY transcription factors and expression analysis of selected genes under abiotic stress in the barley genome. The manuscript is well written with few exceptions. This manuscript can be accepted for publication after incorporation of the following changes.

  1. On page 2, line 85, the authors can provide reference for HMMER software.
  2. Line 89, ‘submitted SMART’ needs to be rephrased.
  3. In page 3, the authors need to provide the references for the different expression data used from the IPK website. Provide web url and reference for IPK website.
  4. Page 5, Figure 2, the labels are hard to read. The text can be improved.
  5. A good phylogenetic analysis has been carried out here. The authors mentioned that the tree was generated using 1000 bootstrap replications. However, the bootstrap information is not shown in the figure 3. Apart from the different groups of WRKY family, can the authors comment on the evolutionary pattern of this family.
  6. In figure 4, the authors have to provide information about the different color shading used here.
  7. In Figure 7, the authors should provide information about the abbreviations used. It will be helpful for the readers.

Author Response

1. On page 2, line 85, the authors can provide reference for HMMER software.

Response: Thanks. We have made a reference to HMMER.

2. Line 89, ‘submitted SMART’ needs to be rephrased.

Response: Thanks. We have changed 'submitted SMART' to”submitted to SMART website”

3. In page 3, the authors need to provide the references for the different expression data used from the IPK website. Provide web url and reference for IPK website.

Response: Thanks. We have attached the address of the barley page of the IPK website, but the expression information is in the pages of each gene, which is not suitable for display. In addition, we are very sorry that we cannot find references to this website.

4. Page 5, Figure 2, the labels are hard to read. The text can be improved.

Response: We have changed the label of Figure 2 to “Figure 2. Collinearity analysis of WRKY genes in Oryza sativa (pink), Brachypodium distachyon (brown), Hordeum vulgare (gray) and Arabidopsis thaliana (blue). Each horizontal bar represents a chromosome. The orthologous WRKY genes were linked using blue curves.”

5. A good phylogenetic analysis has been carried out here. The authors mentioned that the tree was generated using 1000 bootstrap replications. However, the bootstrap information is not shown in the figure 3. Apart from the different groups of WRKY family, can the authors comment on the evolutionary pattern of this family.

Response: Thank you for your suggestion. We show Bootstrap values in Figure 3 using dot sizes.

6.In figure 4, the authors have to provide information about the different color shading used here.

Response: Thank you. We have explained the meanings of the different color shading.

7.In Figure 7, the authors should provide information about the abbreviations used. It will be helpful for the readers.

Response: Thank you. In the label in Figure 7, we have previously shown the abbreviation details of the source of the sample in the second paragraph of the label. In order to avoid misunderstanding, the two paragraphs are now integrated into one paragraph.

Reviewer 3 Report

Dear Editor,

I have read the manuscript  entitled “Genome-Wide Identification of WRKY Gene Family and Expression Analysis Under Abiotic Stress in Barley” submitted by Junjun Zheng et al for publication in Agronomy.

The study has been conducted mainly using an in silico approach to analyze the barley WRKY gene and proteins. Using this approach the authors to carried out the phylogenetic analysis of barley WRKY proteins and  genes. The authors also performed the analysis of the promoter regions. Moreover the paper reports the results on the transcriptional analysis of barley WRKY genes carried out on different barley tissues at different growth stages and subjected to different stress conditions.

In my opinion the manuscript is very well written and results clearly described and well documented with figures and tables. In my opinion the results reported are of interest for the scientific community and for these reasons, the manuscript merits publication.

Author Response

Thank you very much!